# Surveying the Proteome-Wide Landscape of Mitoxantrone and Examining Drug Sensitivity in BRCA1-Deficient Ovarian Cancer Using Quantitative Proteomics

**DOI:** 10.3390/proteomes13040061

**Published:** 2025-11-14

**Authors:** Savanna Wallin, Sneha Pandithar, Sarbjit Singh, Siddhartha Kumar, Amarnath Natarajan, Gloria E. O. Borgstahl, Nicholas Woods

**Affiliations:** 1Eppley Institute for Research in Cancer and Allied Diseases, Fred & Pamela Buffett Cancer Center, University of Nebraska Medical Center, Omaha, NE 68198, USA; savanna.wallin@unmc.edu (S.W.); spandithar@unmc.edu (S.P.); sarbjit.dhami@gmail.com (S.S.); kumar3sd@ucmail.uc.edu (S.K.); nataraah@ucmail.uc.edu (A.N.); 2Division of Pharmaceutical Sciences, College of Pharmacy, University of Cincinnati, Cincinnati, OH 45267, USA

**Keywords:** mitoxantrone, homologous recombination-deficiency, ovarian cancer, BRCA1, DNA repair, proteomics, ribosome biogenesis

## Abstract

**Background**: Mitoxantrone (MX) is regularly used to treat several cancers. Despite its long history in the clinic, recent studies continue to unveil novel protein targets. These targets may contribute to the cytotoxic effects of the drug, as well as potential non-canonical antitumor activity. A better understanding of MX’s cellular targets is required to fully comprehend the molecular consequences of treatment and to interpret MX sensitivity in homologous recombination (HR)-deficient cancer. **Methods**: Here, we evaluated MX activity in HR-deficient UWB1.289 (BRCA1−) ovarian cancer cells and surveyed the binding profile of MX using TMT-labeled quantitative proteomics and chemoproteomics. **Results**: Mass spectrometry (MS) analysis of cellular extracts from MX-treated BRCA1−UWB1.289 cells revealed unique downregulation of pathways instrumental in maintaining genomic stability, including single-strand annealing. Moreover, the BRCA1− cells exhibited a significant upregulation of proteins involved in ribosome biogenesis and RNA processing. Additional MS analyses following affinity-purification using a biotinylated-mitoxantrone probe corroborated these findings, which showed considerable targeting of proteins involved in genome maintenance and RNA processing. **Conclusions**: Our results suggest that an interplay of both canonical and non-canonical MX-antitumor activity overwhelms the BRCA1− UWB1.289 cells. Furthermore, this study characterizes the target landscape of MX, providing insights into off-target effects and MX action in HR-deficient cancer.

## 1. Introduction

The chemotherapeutic mitoxantrone (MX) is routinely used clinically to treat acute lymphoblastic leukemia (ALL), acute myeloid leukemia (AML), hormone-refractory prostate cancer, and forms of relapsing multiple sclerosis [1,2]. It is also used off-label to treat advanced metastatic breast cancer [2]. Generated initially in 1978, it was first FDA-approved in the 1980s to treat AML. MX is the result of drug discovery efforts towards generating a class of therapeutic agents less cytotoxic than their predecessor, doxorubicin. The canonical mechanisms of action associated with MX are topoisomerase II inhibition and DNA intercalation [1,2,3]. Continued investigation of the drug revealed it binds to RNA, single-stranded DNA (ssDNA), and associates with numerous proteins such as methyl-CpG binding domain 2 (MBD2), the serine/threonine kinase PIM1, apurinic/apyrimidinic endonuclease 1 (APE1), histone complexes, and DNA repair protein RAD52, amongst others [2,4,5,6,7,8]. MX induces several forms of DNA damage, including single and double-stranded breaks, adducts, and DNA-protein cross-links. MX traps topoisomerase II on DNA, preventing the resolution of the cleavage complex, leading to highly toxic double-stranded breaks (DSBs) [2].

The ability to cause DNA damage, inhibit topoisomerase II, and induce cell cycle arrest and apoptosis makes it an ideal anti-cancer therapeutic. Despite its use in the clinic, the complete binding profile and molecular consequences of MX treatment are not well understood. In particular, the off-target effects that influence its cytotoxicity and potentially contribute to its anti-proliferative properties remain an active area of investigation, such as mitochondrial dysfunction and transcriptional disruption (Figure 1).

Understanding off-target effects is key to identifying dose-limiting toxicities; for MX, cardiotoxicity is a prominent example, but remains poorly understood [1]. Its broad application across disease types and dose-dependent effects are clues that underscore our limited understanding of MX’s intracellular interactions. Recent evidence suggests that MX is useful in targeting homologous recombination (HR)-deficient cancer [7]. Upwards of 50% of high-grade serous ovarian carcinoma (HGSOC) cases are categorized as deficient in HR, a DNA repair pathway that is instrumental in efficiently repairing double-stranded breaks [9]. Deficiencies in HR are consequences of pathogenic gene mutations that result in inefficient recruitment of DNA repair proteins or impaired loading of RAD51, the effector protein of HR [9,10]. A few recognized gene mutations associated with deficiencies in HR include *BRCA1*, *BRCA2*, *PALB2*, *BARD1*, and *RAD51C/D* paralogs [10,11]. Studies have shown that mutations in these genes increase the risk of ovarian and breast cancer and decrease the efficiency and accuracy of double-stranded break repair [9,10,11]. However, these mutations also elicit tumor-specific dependencies that are targetable by inducing synthetic lethality [9,10,12]. HR-deficient cancer cells are sensitive to DNA-damaging agents and are dependent on alternative DNA repair pathways to maintain genomic stability. This is evidenced by the success of platinum-based therapies and the use of PARP inhibitors in the clinic for HR-deficient patients, although their use is limited by eventual drug resistance [10,13,14].

Previously, we showed that MX can disrupt the protein-protein interaction (PPI) between RPA and RAD52, which is a critical interaction for HR-deficient cells [7]. Additionally, several HR-deficient cell lines showed heightened sensitivity to MX compared to HR-proficient control cells, suggesting a synthetic-lethal relationship, though the mechanism is not defined [7]. To gain a better understanding, we investigated the comprehensive cellular consequences of MX treatment in HR-deficient BRCA1-mutant UWB1.289 (BRCA1−) and HR-proficient (BRCA1+) UWB1.289 ovarian cancer cells using quantitative shotgun proteomics. BRCA1− UWB1.289 cells harbor a mutation in exon 11 of the *BRCA1* gene and a deletion of the wild-type allele that results in a truncated and non-functional BRCA1 protein [15]. Consequently, this pathogenic mutation impairs HR. Amongst several essential functions in DNA repair regulation and response, BRCA1 aids in the recruitment of other DNA repair enzymes, such as BRCA2, which is instrumental in the loading of RAD51 [16,17,18,19,20,21]. A wild-type BRCA1+ UWB1.289 cell line with restored HR function was used as a control. In parallel to the shotgun proteomic experiments, we used a biotinylated-mitoxantrone probe (MXP) and a no-compound control molecule (MXP-NC) to investigate the cellular binding partners of MX in both cell lines. Here, we characterize the similarities and differences in the target landscape of MX using MXP. We also report the global changes in cellular processes because of MX treatment in an HR-deficient and HR-proficient background. Together, these results provide insight into potential influences of MX sensitivity and into a proteome-wide binding profile associated with MX.

## 2. Materials and Methods

### 2.1. Cell Lines and Reagents

BRCA1− parental (CRL-2945) and BRCA1+ (CRL-2946) UWB1.289 cells were purchased from ATCC. Cells were cultured in 50% RPMI 1640 medium (Gibco 11875093), 50% supplemented Mammary Epithelial Growth Medium (PromoCell C-21010), Penicillin/Streptomycin (Gibco 15140122), and 3% final concentration of fetal bovine serum. The selective antibiotic Geneticin (G418 Sulfate) (200 µg/mL) was added to the BRCA1+ UWB1.289 cell medium to maintain the production of functional BRCA1 (Gibco 10131035). Cells were seeded in 100-mm petri dishes and incubated at 37 °C and 5% CO_2_. Unmodified Mitoxantrone was purchased from Sigma Aldrich (St. Louis, MO, USA) and stored in 100% DMSO as 10 mM stocks in −20 °C. MXP and MXP-NC were generated in-house as previously reported, and stored in 100% DMSO as 1000 µM stocks at −20 °C [8].

### 2.2. Whole Cell Lysate Proteomics

For shotgun proteomic experiments, 1 million cells were seeded into 100-mm petri dishes. Mitoxantrone or an equal volume of DMSO was added to the cell medium. Cells were treated at the previously reported EC_50_, 0.2 µM, for 48 h [7]. After treatment, cells were washed three times with 1X PBS, lysed directly on the petri dish using RIPA buffer supplemented with protease and phosphatase inhibitor cocktail. Cell debris was removed after centrifugation at 10,000× *g* for 10 min, and the lysate was aliquoted and stored at −80 °C. Protein levels were assessed using the CB-X assay kit (G-Biosciences, St. Louis, MO, USA), and 100 µg of each sample was reduced (DTT), alkylated (iodoacetamide), and precipitated with acetone. Samples were washed twice with cold 70% ethanol. Samples were pelleted, and then the pellets were digested in 0.1 M EPPS buffer (pH 8.5) with Lys-C (1:50) for 2 h at 37 °C. Then, they were incubated with trypsin (1:40) for 4 h. The digested samples were analyzed using nano-LC-MS and had >93% zero missed cleavages, making them suitable for TMT labeling and analysis by mass spectrometry. Digest samples (100 µg) were labeled with TMTpro reagent 16-plex labels (0.5 mg) and incubated for one hour at room temperature. Labeling was quenched with 5% hydroxylamine. Labeling completeness was quantified by mass spectrometry before combining the samples and removing acetonitrile. The 16-plex sample was further cleaned by desalting (100 mg SepPak C18 SPE columns) and washing with 10% acetonitrile. Extra TMT-label reagent was removed using 0.1% TFA, and the labeled peptides were eluted using 70% acetonitrile, 0.1% TFA. Peptide quantities were determined and samples pooled into a single TMT labeled 16-plex peptide mixture, which was dried using a SpeedVac (Thermo Scientific, Waltham, MA, USA) and reconstituted in high pH buffer (pH 10). The peptide mixture was sub-fractionated using high pH reverse-phase C18 chromatography at pH 10. Subfractions were combined using concatenation into 12 fractions. These 12 fractions were dried and re-suspended in low pH buffer (5% acetonitrile and 0.05% TFA). Each fraction was run for 2.5 h on a C18 Waters CSH column connected to an Orbitrap Eclipse mass spectrometer. Differentially abundant proteins resultant from MX treatment were identified using Proteome Discoverer (Thermo Scientific, Waltham, MA, USA, v3.2) with SEQUEST HT and COMET. The log_2_ protein abundance ratio was calculated by dividing the summed protein abundance quantified across four replicates for one treatment group by the summed protein abundance of another treatment group. Treatment groups (MX vs. DMSO) for either BRCA1− or BRCA1+ UWB1.289 samples were compared. The associated adjusted *p*-values were determined by analysis of variance (ANOVA) using the Benjamini-Hochberg method. Global protein changes upon MX treatment were visualized with a volcano plot using RStudio v4.4.3. Criteria for proteins considered significantly differentially regulated are a |log_2_ fold change| of >1 and an adjusted *p*-value of less than 0.05. The dynamic range of this approach is demonstrated in Appendix A. Principal component analysis plots were generated using an in-house Python (v3) program. Gene enrichment of the differentially abundant proteins was also analyzed using Enrichr [22,23,24]. The Interactome of MXP-isolated proteins and significantly differentially abundant proteins after MX treatment was assessed using STRING (v12.0) and visualized using Cytoscape (v3.10.3) [25,26]. Mass spectrometry analysis was conducted at the Proteomics & Metabolomics Facility (RRID:SCR_021314), Nebraska Center for Biotechnology at the University of Nebraska-Lincoln.

### 2.3. Affinity Purification of MXP Binding Proteins

MXP was utilized in the pulldown with Pierce™ Streptavidin Magnetic Beads (Thermo Scientific #88817). Following the streptavidin magnetic beads manufacturer’s guidelines, the BRCA1− and BRCA1+ UWB1.289 cells were cultured to 90% confluency, washed with ice-cold 1X-PBS (Thermo Scientific #10010023), and lysed with Pierce™ IP-MS Cell Lysis Buffer (Thermo Scientific #1863073) in addition to protease and phosphatase inhibitors. 300 µL of cell lysate was incubated with either 60 µM of MXP, MXP-NC, or DMSO overnight at room temperature on an orbital shaker. After incubation, the streptavidin magnetic beads (0.5 mg per sample) were pre-washed with 1X TBS three times using a magnetic stand. MXP, MXP-NC, or DMSO-incubated lysates were mixed with the pre-washed beads and further incubated at room temperature for one hour on an orbital shaker. The streptavidin-coated beads were separated using a magnetic stand to isolate the unbound cell lysate. The beads were washed three times by resuspending in 1X TBS, followed by magnetic separation to eliminate weak and non-specific binders. Proteins were then eluted from MXP, MXP-NC, or the beads (DMSO) after a short 5-min incubation with the elution buffer: 100 µL of 0.1 mM glycine (pH 2). The beads were separated a final time, and the elution fractions were neutralized using concentrated Tris (pH 8) until reaching a near-neutral pH. The resulting fractions of supernatants, wash steps, and elutions were aliquoted and stored at −80 °C prior to analysis. Pulldowns were repeated in triplicate for MXP, MXP-NC, and DMSO-bead control in both the BRCA1− and BRCA1+ UWB1.289 cell lysates.

### 2.4. Chemoproteomics

After the affinity purification using MXP, elution, and input samples were diluted with 4X reducing buffer and denatured at 95 °C for 5 min. Samples were run into the top of an SDS-PAGE gel (Bolt^TM^ 12% Bis-Tris-Plus) with MES running buffer, and the gel was fixed in a methanol:acetic acid: water mixture (40:10:50). Then, the gel was washed and stained with Coomassie Blue G250 stain overnight. The gel lane was removed, put through a series of washes, and alkylated to remove traces of SDS and stain. Samples were incubated with trypsin overnight at 37 °C. Evaporated digests were resuspended in 5% acetonitrile and 0.05% TFA. Digests were separated (2-h gradient) using a C18 Waters CSH column connected to an Orbitrap Eclipse mass spectrometer. After the chemoproteomics mass spectrometry workup, MS/MS spectra were analyzed using Mascot (Matrix Science, London, UK, v2.7.0). A fragment ion mass tolerance of 0.060 Da and a parent ion tolerance of 15 PPM was used. Potential residue modifications allowed included deamidation, oxidation, and carbamidomethylation. MS/MS-supported peptide and protein identifications were verified using the software Scaffold (Proteome Software Inc., Portland, OR, USA, v5.3.3). The peptide identification probability threshold was set to 80.0%, using the Peptide Prophet algorithm [27]. True protein identification was limited to proteins with at least two identified peptides and greater than 99% identification probability, which was determined using the Protein Prophet algorithm [28,29]. Mass spectrometry analysis was conducted at the Proteomics & Metabolomics Facility (RRID:SCR_021314), Nebraska Center for Biotechnology at the University of Nebraska-Lincoln.

### 2.5. Proteomics Analysis

Identified proteins were exported to Excel from Scaffold via the Samples report. The Samples report was uploaded to the APOSTL Galaxy Server (Moffitt Cancer Center) for data pre-processing, analysis, and visualization [30,31]. All baits were categorized as either BRCA1− or BRCA1+, depending on which lysate they were incubated in. The BRCA1− and BRCA1+ incubated MXP-NC and DMSO replicates were classified as control baits. Pre-processing of the data using APOSTL yielded individual interaction, prey, and bait files. The interaction, prey, and bait files were used for SAINTexpress analysis with the default number of replicates and no virtual controls. Significance Analysis of Interactome (SAINT) or SAINTexpress assigned interaction probability scores and a Bayesian False Discovery Rate (BFDR) to bait-isolated proteins based on spectral counts, compared to designated controls [30,32,33]. High-confidence protein targets were categorized as proteins with a SAINT score equal to or greater than 0.65. A dotplot of SAINT-scored proteins was generated using ProHits-viz [33,34]. Proteins were visualized according to average abundance (minimum: 2) and BFDR (primary: 0.01, secondary: 0.05). APOSTL Interactive Analysis calculated a normalized spectral abundance factor (NSAF) and a CRAPomePCT based on the abundance of common contaminant proteins reported in the CRAPome [31,35]. The threshold for CRAPomePCT was 0.80 (default). Target proteins were visualized using APOSTL with spectra sum and CRAPomePCT displayed. Protein-protein interaction networks of filtered proteins were visualized using STRING (v12.0) at https://string-db.org/ (accessed 28 July 2025), with a confidence score of 0.4. Gene enrichment analysis was executed through Enrichr at https://maayanlab.cloud/Enrichr/ (accessed 28 July 2025) using Reactome Pathways (2024) and Gene Ontology assessment [22,23,24,26].

### 2.6. DEPMAP

The Cancer Dependency Map is an online resource (https://depmap.org/portal/) (accessed 1 August 2025) with a compilation of multi-omic project datasets [36]. This repository gives access to data associated with the Cancer Cell Line Encyclopedia (CCLE), RNA interference (RNAi) studies [Demeter2], CRISPR screens [Chronos], amongst several other screens [37,38,39]. This study utilized the publicly reported RNAi and CRISPR co-dependency data for both TOP2A and BRCA1 [Demeter2, Chronos]. The top 100 co-dependencies for each gene were compared with MXP-isolated cell targets to assess our dataset for any underlying gene relationships that could influence cell viability after MX treatment in HR-deficient cells. Each dataset yielded a list of genes and the corresponding Pearson correlation. The correlation represents the strength of the relationship between genes. For instance, the effect of diminishing one gene, such as cell viability, correlates with the impact of a decrease in the other. Co-dependency analyses facilitate the identification of likely functionally related genes (positive correlation) and oppositely regulated genes (negative correlation) across various cancer types.

### 2.7. ProteinSimple (Bio-Techne) Peggy Sue Simple Western (Supplemental)

RAD52 is a low-abundance protein that was not detected using mass spectrometry in either of the proteomic experiments. RAD52 abundance was measured using the Peggy Sue Simple Western in both BRCA1− and BRCA1+ UWB1.289 cell lysates, as well as in the affinity purification experiments for MXP, MXP-NC, or DMSO-elution fractions. The protein concentration of the prepared lysates and elution fractions, as described in the Pulldown Assay section, was estimated using the Pierce™ BCA Protein Assay kit (Thermo Scientific 23225). A ProteinSimple Size Separation Module [12–230 kDa] (#SM-S001) was used to ready all samples, according to the manufacturer’s recommendations. Sample buffer (0.1X) and fluorescent standard master mix (FSMM) were prepared according to guidelines and used to dilute samples to their final concentrations. The lysate samples were diluted to 0.25 mg/mL, and the MXP-elution fractions were concentrated using a SpeedVac and made to a final concentration of approximately 0.04 mg/mL. Samples were vortexed and denatured on a heat block for 5 min at 95 °C. The primary RAD52 antibody (Protein Tech #28045-1-AP) was diluted using module-provided antibody diluent (1:50). A ProteinSimple Anti-Rabbit Detection Module (#DM-001) containing the chemiluminescent detection reagent (Luminol-S, Peroxide), anti-rabbit secondary antibody, and streptavidin-HRP were formulated as suggested. Prepared samples, including a biotinylated ladder standard, antibody diluent (blocking), primary and secondary antibodies, separation matrix, stacking matrix, and detection reagent, were loaded into the 384-well assay plate. Compass Software v5.0.1 (Simple Western) was utilized for automated assay development and data analysis. Experiments were set up to run 1–2 cycles with cell lysates (BRCA1− or BRCA1+) matched with corresponding MXP, MXP-NC, or DMSO elution fractions. The internal standards in each sample were verified by comparison to the molecular weight standards in the ladder.

## 3. Results and Discussion

### 3.1. Comprehensive Molecular Characterization of MX Treatment in BRCA1− UWB1.289 Cells Using Quantitative Shotgun Proteomics

This study aimed to employ a multi-pronged proteomics approach to better define the molecular targets of MX and to examine the differences in cellular response that influence MX sensitivity in BRCA1− UWB1.289 ovarian cancer cells (Figure 2A,B). Changes in protein abundance were quantified by evaluating the log_2_ protein abundance ratio, which was calculated as the ratio of summed abundances for identified proteins between treatment groups (MX vs. DMSO) for either BRCA1− or BRCA1+ UWB1.289 samples. The principal component analysis (PCA) following the shotgun proteomic workflow is shown in Figure 2C. First, BRCA1− and BRCA1+ UWB1.289 cells were treated with 0.2 µM of MX or DMSO for 48 h, and protein abundance was measured using quantitative TMT-labeled mass spectrometry. MX treatment induced pronounced differences in cellular responses between BRCA1− and BRCA1+ cells. Differentially abundant proteins were defined as proteins with a log_2_ fold change greater than one and an adjusted *p*-value of less than or equal to 0.05 (Figure 2D).

Notably, Reactome analysis of the downregulated proteins upon MX treatment in BRCA1− cells revealed a significant decrease in proteins active in double-stranded break repair and pro-apoptotic pathways (Figure 3A). One of these pathways included single strand annealing (SSA), which is a lifeline to HR-deficient cells to maintain genomic stability [40]. Analysis of the upregulated proteins exposed a striking upregulation of proteins involved in RNA metabolism and rRNA processing (Figure 3A). It is widely known that cancer cells largely depend on increased ribosome biogenesis; thus, an upregulation of proteins integral to rRNA modification and rRNA processing machinery is likely a compensatory response to cellular stress [41,42]. It is recognized that MX obstructs the early steps of transcription primarily by intercalation, and it is also known to modulate DNA topology, which may impede RNA transcription by decreasing DNA access for RNA polymerases or transcription factors [43,44]. Moreover, TOP2A has been shown to enable RNA polymerase I activity via assisting in the assembly of the pre-initiation complex through topological changes in DNA [45]. Hence, MX inhibits early rRNA transcription steps through DNA intercalation and TOP2A trapping, thereby affecting DNA and RNA synthesis. However, it is unclear how the inhibition of early transcription events significantly affects rRNA processing in BRCA1− cells. The resulting upregulation of these proteins after MX treatment in BRCA1− cells suggests global disruptions in rRNA metabolism, particularly in rRNA modification and maturation, not solely the early steps of RNA transcription, and supports that BRCA1 dysfunction impacts RNA metabolism.

Similar to the BRCA1− UWB1.289 cells, the BRCA1+ cells downregulated proteins associated with pro-apoptotic pathways (Figure 3B). However, in contrast, they had a substantial upregulation in proteins linked to cell cycle transition and cell growth (Figure 3B), indicating that the BRCA1+ cells were better equipped at overcoming MX-induced cellular stress. Figure 3C illustrates the differences in protein abundance between DMSO-treated BRCA1− and BRCA1+ UWB1.289 cells, supporting that the disruption in DNA repair and disturbances in RNA metabolism are consequences of MX treatment in BRCA1−cells.

### 3.2. Exploring the Target Landscape of MX in BRCA1− and BRCA1+ UWB1.289 Ovarian Cancer Cells Using MXP

To explore the target landscape, we used MXP in a pulldown assay to profile binding partners of MX in BRCA1− and BRCA1+ cells (Figure 4A). After mass spectrometry analysis, a total of 3659 and 3856 proteins were identified across all the baits (MXP, MXP-NC, and DMSO) and inputs for the BRCA1+ and BRCA1− samples, respectively. The required criteria for protein identification in Scaffold were at least two identified peptides and a 95% peptide and protein threshold.

The MXP-isolated proteins were further filtered according to their quantified abundance compared to MXP-NC and DMSO-isolated proteins as negative controls. The Significance Analysis of INTeractome (SAINT) algorithm score was assigned using Automated Processing of SAINT Templated Layouts (APOSTL) accessed through Galaxy [30,32]. SAINT scores predict the likelihood of true interactions based on spectral abundance across baits and controls [32]. Identified proteins with a SAINT score of less than 0.65 were excluded from downstream analysis. Additionally, proteins were filtered using the Contaminant Repository for Affinity Purification (CRAPome) to reduce highly abundant background proteins that may bias the results. The CRAPome is a database of common background contaminant proteins identified by other negative control affinity purification mass spectrometry (AP-MS) experiments [35]. A CRAPome filter cutoff to distinguish high-confidence interactions was assigned as a CRAPomePCT of 0.80 or 80%. Proteins that met the SAINT score and CRAPomePCT filters were included in further analysis (Figure 4B).

Overall, a total of 148 proteins met these thresholds (Figure 4C). A considerable overlap of 101 proteins was observed between targets identified from the BRCA1+ lysate and proteins identified from the BRCA1− cell lysate. Yet 41 unique targets were only associated with the BRCA1− cells (Figure 4D). Of the overlapping proteins identified, the spectral abundance varied between BRCA1− and BRCA1+ samples (Figure 4E). For instance, several proteins involved in DNA replication and early DNA damage response had increased average or unique spectra detected in the BRCA1− samples, such as RFC1, TOP2A, MDC1, RFC4, NSD2, and RIF1 (Figure 4F). To assess the level of enrichment, the enrichment factor (EF) was calculated by taking the ratio of the normalized spectral abundance factor (NSAF) values for proteins isolated by MXP and proteins identified from matched input lysate samples (Table 1). The isolation or enrichment of RFC1, TOP2A, MDC1, RFC4, NSD2, and RIF1 by MXP is intriguing because of their particular roles in advancing DNA damage repair [46,47,48,49,50]. Although these particular proteins did not meet the log2 fold change threshold in the shotgun proteomic analysis, the normalized abundances significantly shifted after MX treatment (Appendix A). Naturally, MX will induce DNA damage and inhibit topoisomerase II function, which directly and indirectly affects DNA replication, repair, and transcription [51,52]. HR-deficient cells are at a disadvantage because they inefficiently repair DSBs; this liability, coupled with MX-induced DNA damage, forces HR-deficient cells to use alternative pathways to maintain genomic stability, such as non-homologous end-joining (NHEJ) or SSA [9,10,12]. MDC1 and NSD2 influence the recruitment of other DNA repair factors, while the RFC complex (containing RFC1 and RFC4) aids in the direct loading of PCNA, which cooperates with polymerases to replicate or repair DNA [46,48,50]. In the context of HR-deficiency, RIF1 is a critical factor that aids in shielding the DNA strands from end-resection [49]. This inhibition of resection is required for NHEJ to progress, otherwise end-resection promotes HR and SSA.

Interestingly, the shotgun proteomic studies revealed decreased SSA in BRCA1−cells, suggesting influences on repair pathway choice after MX treatment. Moreover, our previous work revealed that MX indeed decreased SSA levels in HR-proficient cells as well, potentially mediated through RAD52-inhibition, but more mechanistic studies are needed to clarify [7]. To this end, we have also previously shown that MXP can isolate and interact with RAD52 in cell extract, although it was not initially identified in the mass spectrometry analysis presented here [8]. Further examination of the MXP-elution fraction using Simple Western did identify RAD52 (Appendix A). Additional surface plasmon resonance (SPR) analysis with MXP immobilized and RAD52 (168–306) used as an analyte, which contains the RPA-binding domain, yielded a K_D_ of approximately 4 µM (Appendix A). Taken together, the BRCA1− cells depend on distinct DNA repair enzymes to maintain genomic stability, and our chemoproteomic results suggest that MXP interacts with early DNA damage responders that may influence and disrupt replication and downstream repair.

We next examined the interactome of MXP-isolated proteins from both BRCA1− and BRCA1+ cells to identify protein complexes and PPIs that potentially orchestrate MX-cytotoxicity (Figure 5A). The STRING network illustrated dense connectivity and overlaps between clusters (dashed lines), demonstrating a wide range of PPIs that are involved in adjacent functional pathways. STRING and Biological Networks Gene Ontology (BINGO) analysis revealed MXP-isolated cellular targets were significantly involved in RNA processing, ribosomal biogenesis, chromatin organization, mitochondrial translation and metabolism, aside from DNA repair (Figure 5A,B). Other research groups have highlighted various chemotherapeutic agents and their abilities to disrupt ribosomal biogenesis at different stages [51,53,54]. In terms of MX, potent rRNA transcription inhibition has been reported at clinical doses [51]. Presumably, the canonical targeting of topoisomerase II and intercalation of DNA inhibits the transcription of rDNA, as mentioned before. However, our results demonstrated that MXP isolated subunits of RNA polymerase I (POLR1G and POLR1E) as well as dozens of proteins involved in rRNA processing, which are integral for the transcription of rRNA genes and necessary for ribosome assembly.

It should be noted that the clusters of RNA metabolism-associated proteins and mitochondrial translation-associated proteins are not necessarily all interactors of MXP, but are instead likely isolated by MXP within tightly bound protein complexes. This is one limitation of this approach. Affinity purification techniques, such as our MXP affinity purification, cannot definitively distinguish between direct from indirect protein-protein interactions. From the STRING analysis in Figure 5A, we observe clusters of functionally related proteins with documented physical interactions. This is evidence that MXP is isolating known protein complexes, but the approach does not delineate which protein(s) in the complex MXP directly physically interacts. However, the results provide insight into pathways that are targeted by MXP through the complexes that are isolated. Even so, MXP-isolation of RNA polymerase I subunits and other rRNA processing machinery provides evidence of a supplementary layer of transcriptional inhibition as well as disruption of downstream ribosome biogenesis that is congruent with our shotgun proteomic results and coincides with what is reported in the literature [43,44,51,52]. In addition, cell component analysis showed substantial enrichment in proteins associated with mitochondrial ribosomes or components of ribosomes, as well as an enrichment in nucleolar proteins (Appendix A). Consistent with these findings, the cellular targets of MXP that were specifically identified in BRCA1− cell lysate are involved in ribosomal biogenesis and RNA maturation. (Appendix A). MXP-interaction with subunits of RNA polymerase I and other ribosomal processing components may potentially compound with the disruptions in DNA and RNA synthesis, but further studies are required to assess the strength of these interactions.

Other noteworthy clusters in Figure 5A include a dense grouping of proteins required for mitochondrial translation, as well as subunits involved in ATP synthase. Previous research has demonstrated that MX-induced cardiotoxicity is, in part, caused by mitochondrial dysfunction and energetic imbalance [1,55,56]. A large contributor to the energetic imbalance induced by MX is the decrease in oxidative phosphorylation, fatty acid oxidation, the citric acid cycle, and overall ATP production [56]. The mitochondrial translation-related proteins and ATP synthase components isolated by MXP offer further evidence that aligns with existing literature. Overall, we observed MXP interacting with the canonical target of MX (TOP2A) and proteins involved in early stages of DNA repair. We also identified proteins essential for the various stages of ribosome biogenesis, including early rRNA transcription, rRNA processing, and ribosome assembly, as well as components required for mitochondrial translation. Together, these targets potentially underlie the compounding mechanisms of ribosomal and mitochondrial dysfunction associated with MX.

### 3.3. Identifying Essential Gene Relationships That Influence Cellular Response and May Incite Synthetic Lethal Interactions upon MX Treatment

To further examine the interactome of MX and its global consequences, we integrated the findings from the shotgun proteomic and chemoproteomic studies and used the DEPMAP portal for a cross-analysis to evaluate essential gene relationships. First, we identified overlaps between datasets to pinpoint MXP targets that were also differentially regulated after MX treatment (Figure 6A), representing high-confidence targets linked to a cellular response. In total, four proteins were identified, including TOP2A, cytoskeleton-associated protein 2 (CKAP2), Histone H1.0 (H1-0), and fibroblast growth factor-binding protein 1 (FGFBP1) (Figure 6B). Three of these targets provide validation for MXP as a representative probe of MX, and one represents a potential novel target of MX. As mentioned before, TOP2A is the widely acknowledged target of MX [2]. Moreover, the isolation of H1-0 using MXP corresponds with previous evidence that MX interacts directly with histone proteins, supporting activities of MX at the chromatin level [57,58]. Likewise, FGFBP1 also appears in the literature related to MX. Interestingly, it was found to be differentially upregulated in MX-resistant metastatic prostate cancer tumor samples [59]. FGFBP1 reportedly influences cancer progression and enables metastasis in several cancer types, and its increased expression is associated with aggressive disease and poor prognosis [60,61,62,63,64]. In our studies, MXP isolated FGFBP1, and MX treatment appeared to downregulate its abundance in both BRCA1− and BRCA1+ cells. This supports previous observations of MX-induced differential regulation of FGFBP1 and its status as an anti-cancer target. Lastly, although no reports of direct interactions between MX and CKAP2 appear in the literature to our knowledge, MX is known to alter microtubule dynamics [65]. CKAP2 is involved in chromosomal segregation and stabilization of microtubules and is also co-expressed with TOP2A, as they both colocalize to chromatin and are critical for mitosis [66,67,68]. As such, our STRING analysis demonstrated that CKAP2 and TOP2A are functionally related, but they are not known to physically interact. In fact, according to the STRING database, there are no direct physical interactions known between TOP2A, CKAP2, H1-0, and FGFBP1, suggesting that these targets interact directly with MXP, rather than through protein complex isolation. Nonetheless, CKAP2’s interaction with MXP and its differential abundance potentially suggests a novel MX target, though more studies are needed.

Based on our proteomic findings, we subsequently surveyed whether any MXP-isolated targets coordinated to essential genes reported in the DEPMAP portal [36]. MXP targets were compared to the CRISPR and RNA interference (RNAi) co-dependency gene sets published for TOP2A and BRCA1. These screens used CRISPR or RNAi to interrogate gene relationships. The resulting Pearson correlation for each gene represents the strength of co-dependency for the gene of interest. Strong positive correlations generally represent functionally related proteins, or that the essentiality of the genes is similar. For instance, the CRISPR screen for *BRCA1* gene co-dependencies yielded *BARD1*, *PALB2*, and *BRCA2* as the top three genes with strong Pearson correlations of approximately 0.54, 0.32, and 0.31, respectively. All three of these proteins are known to be directly associated with BRCA1, and perturbation of any of these genes can also result in HR-deficiency [10,16,21,69].

Since our study employed a well-known TOP2A poison and BRCA1-mutant ovarian cancer cells, we were interested in the gene co-dependencies reported for TOP2A and BRCA1 that could influence MX sensitivity seen in BRCA1− cells. After comparing the MXP targets to the DEPMAP co-dependent gene lists for both TOP2A and BRCA1, three proteins of interest overlapped. The three MXP-isolated proteins, AHCTF1, NOP10, and DHX37, overlapped with TOP2A co-dependent genes, whereas no MXP-isolated proteins overlapped with the BRCA1 co-dependent genes. Therefore, only the overlap between TOP2A co-dependencies and MXP-isolated targets is shown in Figure 6C. The reported Pearson correlations and dataset origins are reported in Figure 6D. Analogous to our previous findings, NOP10 and DHX37 are integral for ribosome biogenesis and rRNA processing, while AHCTF1 is involved in nuclear pore assembly [70,71,72,73]. Typically, a strong positive correlation is considered to be near 0.5, so caution must be exercised when interpreting this overlap. However, the identification of proteins related to RNA metabolism critical to ribosome biogenesis in three separate analyses merits additional scrutiny. Interestingly, DHX37 was uniquely identified in BRCA1− cells, while NOP10 and AHCTF1 were isolated by MXP in both BRCA1− and BRCA1+ cells. Although DHX37 did not meet the fold-change threshold required to be termed ‘differentially abundant’ after MX treatment, the average normalized abundance was significantly decreased after MX treatment in both BRCA1− and BRCA1+ cells (Figure 6E). DHX37 is an RNA helicase required for the assembly of pre-ribosomal complexes, and depletion of DHX37 triggers the surveillance pathway responsible for degrading rRNA [71,73]. DHX37 is also important for the resolution of R-loops, which are DNA:RNA hybrid structures and can be byproducts of transcription [74]. Ultimately, these structures lead to DNA damage and replication stress if left unresolved. Markedly, research has shown that BRCA1 also functions in R-loop-associated DNA repair and that depletion or mutation of BRCA1 is associated with an accumulation of R-loops [75]. From these results, targeting DHX37 in parallel with TOP2A in BRCA1− cells could be a promising combination. Disturbing rRNA processing and R-loop maintenance have been highlighted as a promising therapeutic approach, especially in HR-deficient tumors. However, additional validation is needed to clarify how MX directly or indirectly alters ribosome biogenesis and R-loop maintenance in BRCA1-associated cancer.

Ultimately, our results support that the BRCA1− cells are vulnerable to disturbances in ribosome biogenesis and that broad targeting of DNA repair and RNA metabolism may potentiate synthetic lethal interactions. When interpreting these results, it is important to consider RNA processing and its emerging role in DNA damage repair, as well as BRCA1’s influence on this process [76]. BRCA1 is functionally and physically associated with several MXP-isolated targets (Appendix A). Aside from BRCA1’s role in HR and recruiting other DNA repair factors, it is also known to function in transcriptional regulation and mRNA splicing in response to DNA damage, as well as R-loop-related DNA repair [17,18,19,20,21]. Research has shown that BRCA1-dependent HR repair is enabled by interaction with pre-rRNA and that BRCA1 influences the regulation of RNA Polymerase I, II, and III, thereby affecting downstream ribosome biogenesis [17,19,21,77]. From our results, we observe a decrease in proteins involved in important DNA repair mechanisms following MX treatment, in addition to the identification of early DNA damage responders that influence DNA resection by MXP. We also observed a significant increase in proteins associated with RNA metabolism and ribosome biogenesis in HR-deficient cells, a compensatory response consistent with broad targeting of RNA transcription and processing machinery.

Though the exact influence of BRCA1-deficiency on ribosome dysfunction remains unclear, it is evident that MX-induced DNA damage coupled with MX-dependent inhibition of transcription and disruption of ribosome biogenesis is potent in BRCA1− cells. This outcome is consistent with the effects seen with similar anticancer compounds. Such examples include other intercalating and DNA-damaging agents that have been found to elicit levels of ribosomal stress that contribute to their cytotoxicity [51,53]. The inhibitor CX-5461 is a notable example, though it was initially considered an RNA polymerase I inhibitor, continued investigation revealed it also inhibits topoisomerase II, causes DNA damage, and stabilizes G-quadruplex structures in DNA, activities akin to MX [78,79,80,81,82,83]. CX-5461 is currently undergoing early-phase clinical trials and has shown promise in inducing synthetic lethality in BRCA1 and BRCA2 mutant cancers [78,81,82]. Taken together, the combination of topoisomerase II inhibition, DNA damage, and its disrupted repair, along with the broad targeting of components necessary for ribosomal biogenesis, exploits a secondary vulnerability in BRCA1-deficient cancer (Figure 7).

## 4. Limitations and Future Directions

Although this work provides clarity to the understudied proteome-wide binding profile of MX, it is important to note the limitations of this study that may impact our findings. Our study aimed to explore the differences in MX sensitivity in HR-proficient and deficient cancer cells by examining the cellular targets and molecular consequences of MX. However, there is a wide variation of mutations that disrupt HR, other than BRCA1. This system provides a workflow for deciphering differences, but evaluation of other HR-deficient models is required to validate these findings. Secondly, the proteome is complex and comprises distinct proteoforms. This study does not account for the varying proteoforms and focuses only on canonical proteins. Additionally, the lysate preparation and mass spectrometry workup may preclude the identification of biologically important low-abundant proteins. Future studies should examine additional HR-deficient models and potentially use more quantitative studies for MXP-target validation. Also, it would be appropriate to classify high-affinity binding partners due to the dose-dependent effects seen with MX, which can be supported with competition studies using MX in combination with MXP. Lastly, garnering evidence that supports synergistic targeting of DNA repair and RNA metabolism would strengthen the translation of this work.

## 5. Conclusions

The study presented here employed a multi-pronged proteomics approach to explore the consequential cellular response to MX treatment and the expansive binding profile of mitoxantrone to gain insight into MX sensitivity in BRCA1-deficient cancer. MX treatment in BRCA1− cells revealed a decrease in SSA and a prominent upregulation of proteins associated with ribosome biogenesis. MXP-isolated proteins substantiated these observations with the identification of the canonical target of MX (TOP2A) while illuminating the broad targeting of proteins associated with DNA damage repair, and ribosome biogenesis. Our results suggest that MX perturbs DNA repair and RNA metabolism, which is an advantage in BRCA1− cells. Ultimately, this work provides insight into the broad selectivity of MX that underlies its general cytotoxicity and exposes an additional vulnerability in BRCA1− UWB1.289 cancer cells.

## Figures and Tables

**Figure 1 proteomes-13-00061-f001:**
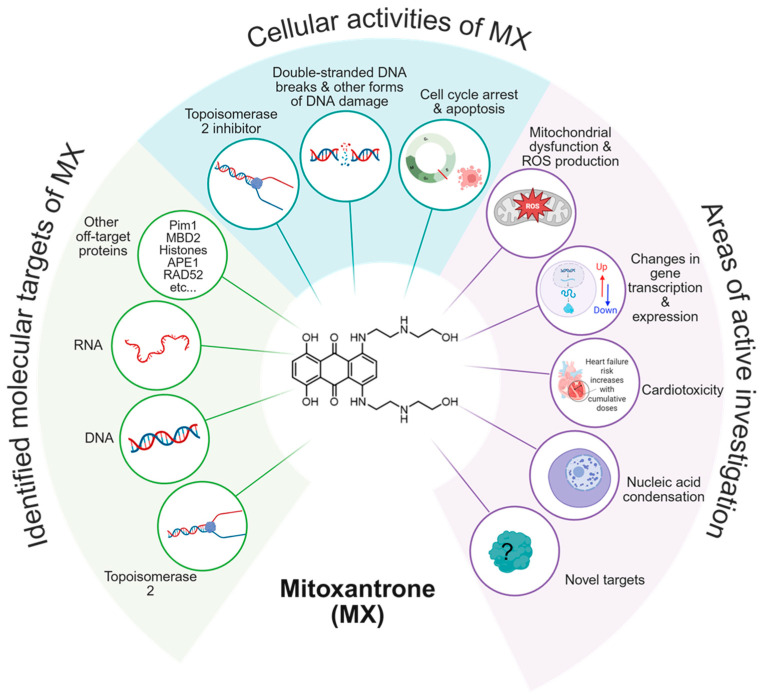
The known targets, cellular activities, and growing areas of interest associated with MX.

**Figure 2 proteomes-13-00061-f002:**
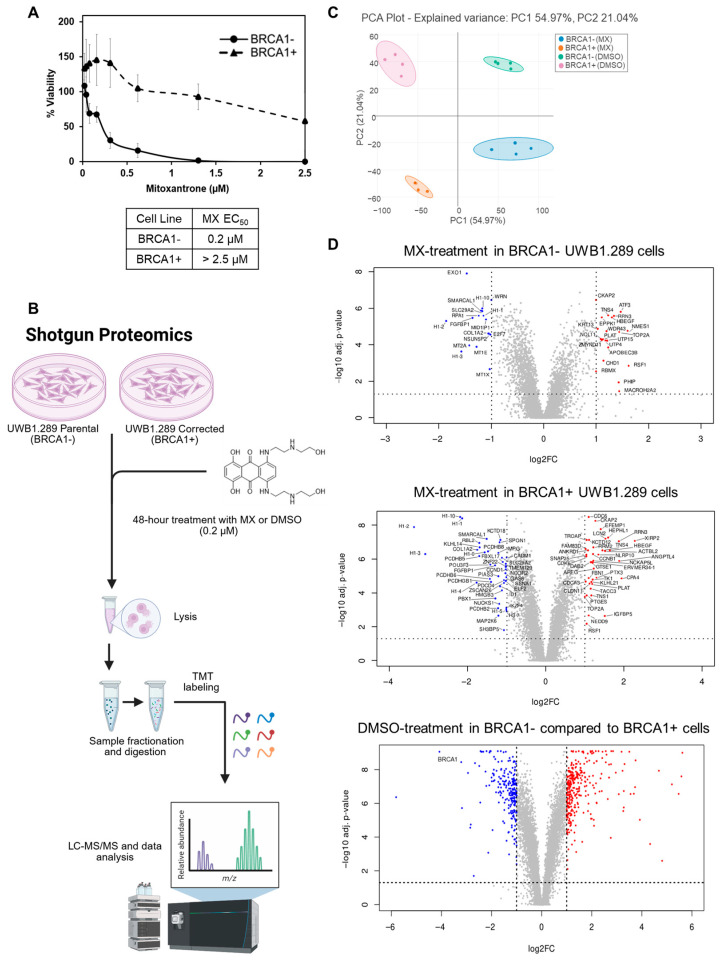
Using shotgun proteomics to assess MX action in homologous recombination (HR)-deficient cells. (**A**) Cell viability of HR-deficient BRCA1− UWB1.289 ovarian cancer cells compared to HR-proficient BRCA1+ UWB1.289 ovarian cancer cells after treatment with MX. The EC_50_ values were quantified and noted in the table below the viability chart. The BRCA1− cells were significantly more sensitive to MX treatment. (**B**) The workflow illustrates the evaluation of the global consequences of MX treatment through 16-plex TMT-labeled shotgun proteomics in HR-deficient and proficient UWB1.289 cells. Cells were treated for 48 h with 0.2 µM MX (or an equal volume of DMSO), and protein abundance was assessed using quantitative mass spectrometry. (**C**) PCA plot depicting variance in BRCA1− or BRCA1+ MX-treated samples. Each treatment was completed in biological quadruplicate. The color-coded circles represent the 95% confidence interval for each group of samples. (**D**) Volcano plots were generated using RStudio, which illustrate significant global protein changes in MX-treated BRCA1− and BRCA1+ UWB1.289 cells. Significantly upregulated (red) or downregulated (blue) proteins are highlighted and labeled. The dashed lines signify the significance thresholds: log_2_ fold change (≤−1 or ≥1), or −log_10_ adjusted *p*-value (0.05). Differences in the DMSO-treatment condition between BRCA1− and BRCA1+ cells were quantified in parallel to highlight the MX-associated effects. Due to the number of significant proteins, only BRCA1 was labeled.

**Figure 3 proteomes-13-00061-f003:**
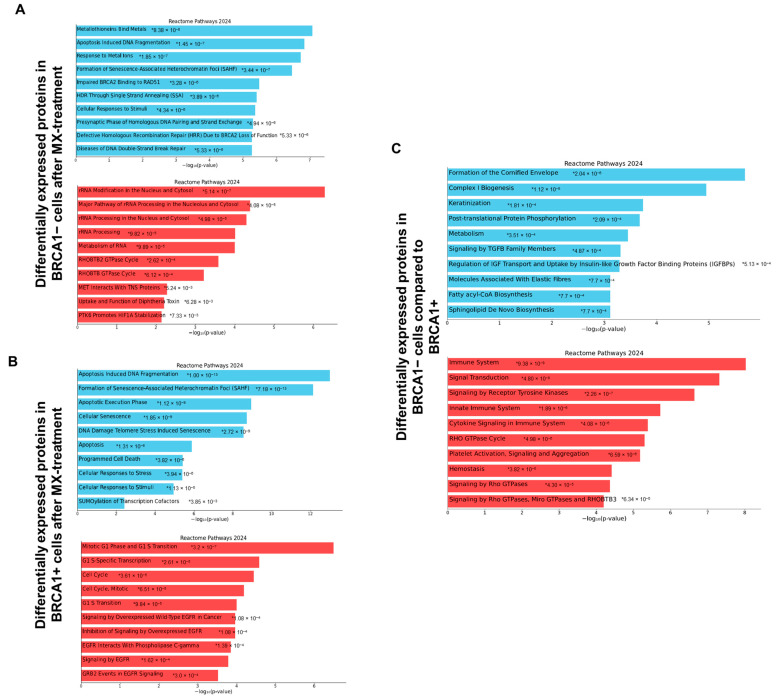
Reactome pathways analysis of MX treatment in BRCA1− and BRCA1+ cells. Reactome pathways analysis using Enrichr (accessed via Appyter) showed a number of enriched pathways that were perturbed upon MX treatment in BRCA1− (**A**) and BRCA1+ (**B**) cells. The *p*-values associated with each enriched pathway are located in the bar graphs, noted by an asterisk. To distinguish between cell differences and MX-driven changes, the DMSO treatments were also assessed (**C**). MX treatment in BRCA1− cells causes a downregulation in DNA repair pathways and a stark increase in rRNA processing pathways, suggesting ribosomal stress.

**Figure 4 proteomes-13-00061-f004:**
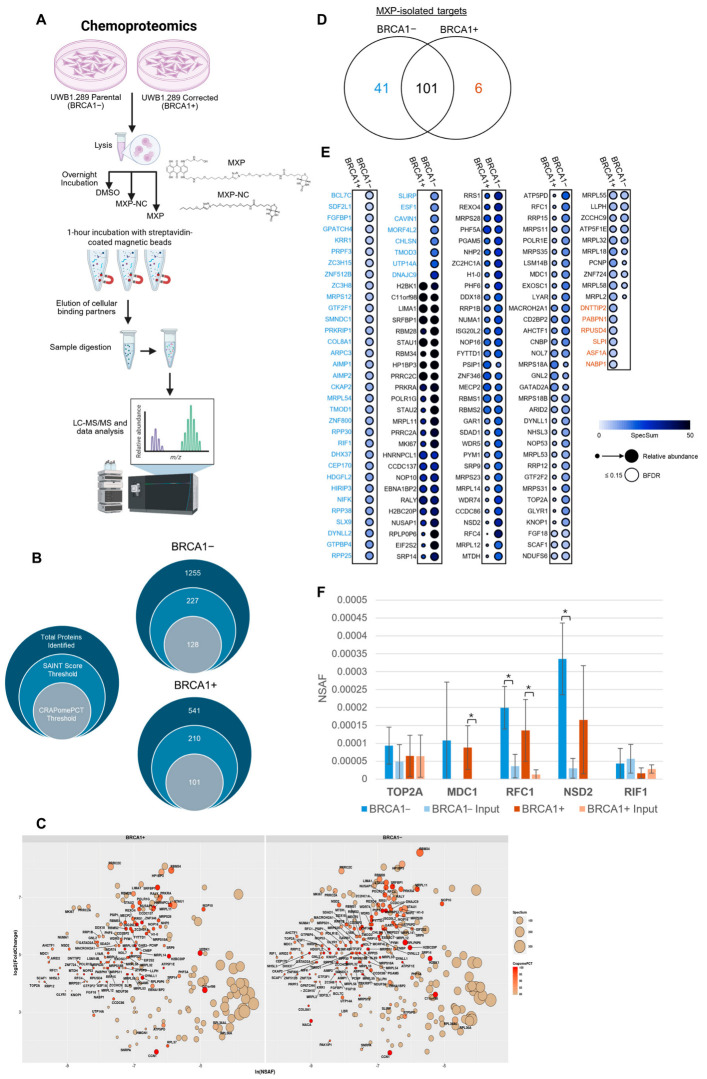
Using chemoproteomics to survey the proteome-wide landscape of MX. (**A**) The chemoproteomic workflow used to identify the cellular targets of MX using MXP. (**B**) MXP-isolated proteins identified in either the BRCA1− or BRCA1+ UWB1.289 cell lysates were first filtered using Scaffold. Filter requirements were set to a minimum of 2 peptides identified at 95% Protein and Peptide thresholds. Identified proteins exported from Scaffold were further filtered by Significance Analysis of INTeractome (SAINT) algorithm score (0.65) and CRAPomePCT (0.80) using APOSTL (accessed via Galaxy). (**C**) MXP-isolated proteins that met the SAINT score and CRAPomePCT thresholds are shown relative to their fold change and normalized spectral abundance factor (NSAF) compared to MXP-NC and DMSO controls. The overall spectral sum is represented by the size of the circle. A larger version is available in the supplement (Appendix A). (**D**) Venn diagram showing the overlap of SAINT-filtered (0.65) and CRAPomePCT (0.80)-filtered proteins identified from BRCA1− and BRCA1+ UWB1.289 cell extract. (**E**) Dot plot generated using ProVizHits illustrating the average abundance quantified for each protein that met filtering thresholds. The size and color gradient (blue to black) of the circles represent the abundance relative to the bait controls (MXP-NC and DMSO). The Bayesian false discovery rate (BFDR) was calculated using APOSTL (accessed via Galaxy). The black border represents a BFDR of less than or equal to 0.15. (**F**) A bar chart depicting the normalized spectral abundance factor (NSAF) values of DNA repair proteins isolated by MXP. The NSAF values of proteins detected in the MXP pulldowns were compared to the input lysate samples from either BRCA1− and BRCA1+ cells by an unpaired *t*-test. Significant differences were defined as a *p*-value of less than 0.05 and are noted by an asterisk. The standard deviation across three replicates is shown.

**Figure 5 proteomes-13-00061-f005:**
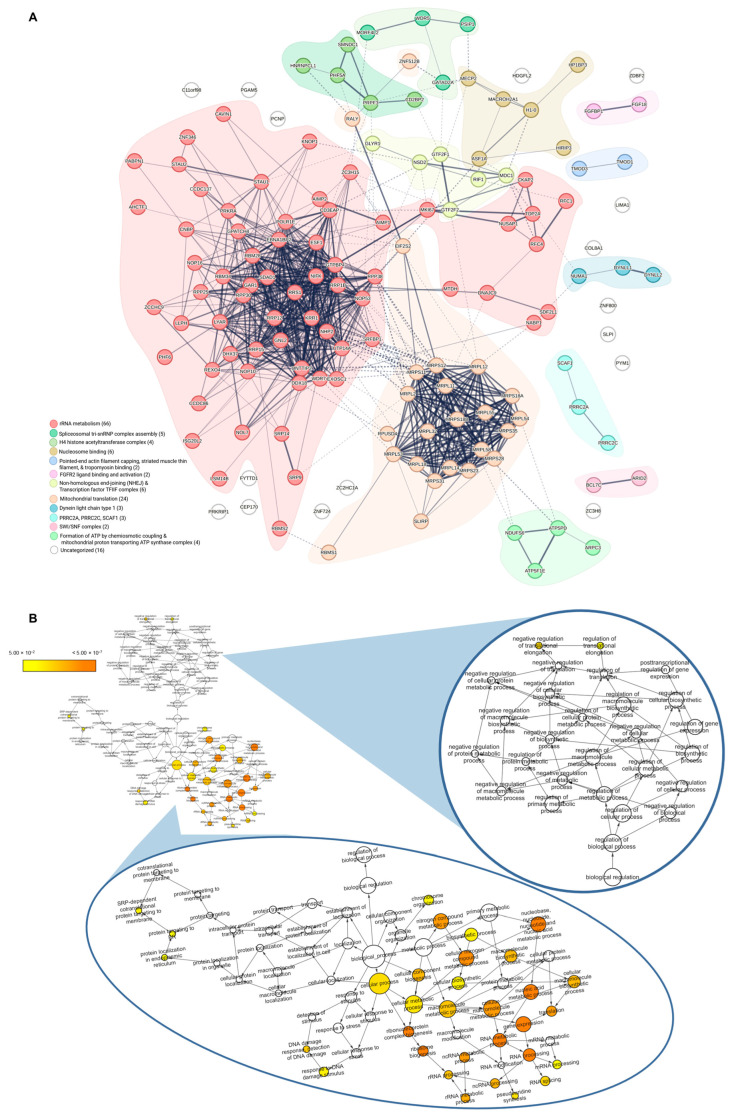
The biological pathways and processes associated with the proteins isolated in BRCA1− cells by MXP are largely associated with DNA repair and RNA metabolism. (**A**) The network of filtered-MXP targets was created using STRING and was clustered using k-means clustering (12 clusters). Solid lines indicate known interactions between nodes, and the intensity of the line increases with the confidence of the interaction. Dashed lines represent known interactions between clusters. Clusters are grouped functionally and physically, and STRING assigns a primary, and occasionally a secondary and tertiary description that best represents the cluster. The descriptions are based on gene ontology categories and are color-coded in the legend below the network. The number of genes categorized in each cluster is found in parentheses following the description. (**B**) Biological Networks Gene Ontology Tool (BINGO) analysis of MXP-isolated proteins using Cytoscape. This analysis gives a more in-depth assessment of the RNA metabolism processes associated with MXP-isolated proteins. The significant level of the enrichment is denoted by the yellow to orange gradient map (*p* ≤ 0.05).

**Figure 6 proteomes-13-00061-f006:**
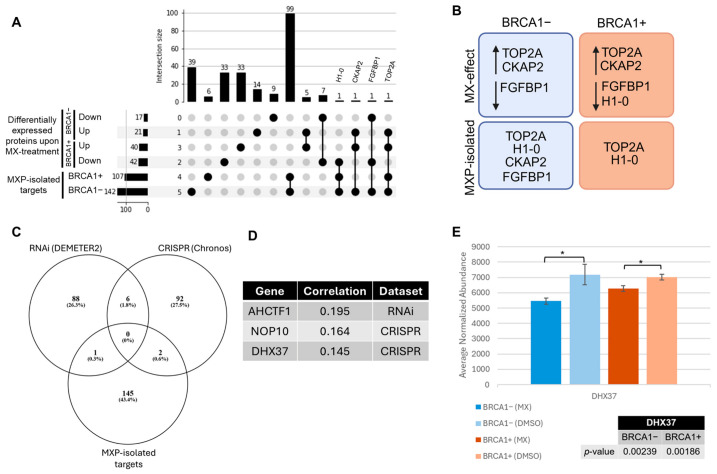
Comparison of chemoproteomic, shotgun proteomic, and DEPMAP Gene Co-dependency datasets. (**A**) The upset plot was generated using the Appyter set comparison tool by the Maayan Lab. It shows the overlap of significant proteins identified in both proteomic approaches. The numbers of proteins identified in either proteomic workflow are categorized based on whether they were identified in BRCA1− or BRCA1+ cells. For the differentially abundant proteins upon MX treatment, downregulation or upregulation are indicated by “Down” or “Up”, respectively in either cell line. (**B**) An illustration showing the four proteins that appeared in both datasets: TOP2A, CKAP2, FGFBP1, and H1-0. Here, the significant associated changes in abundance after MX treatment and the isolation by MXP are shown with respect to the BRCA1− or BRCA1+ UWB1.289 cells. (**C**) Venn Diagram illustrating the overlap between the TOP2A RNAi (DEMETER2) and CRISPR (Chronos) co-dependency datasets from the DEPMAP portal, with the identified targets of MXP from either BRCA1− or BRCA1+ cells. (**D**) A table of the names of the overlapping proteins from (**C**) with their corresponding Pearson correlations from the co-dependency gene set list reported for TOP2A. (**E**) The average normalized abundance was compared between MX and DMSO-treated BRCA1− and BRCA1+ cells using an unpaired *t*-test, assuming equal variance. A *p*-value less than or equal to 0.05 was considered significant and is noted by an asterisk.

**Figure 7 proteomes-13-00061-f007:**
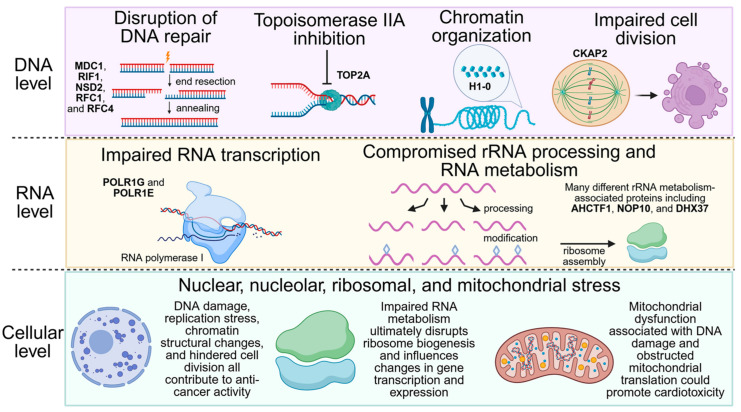
MX effect in BRCA1− UWB1.289 cells.

**Table 1 proteomes-13-00061-t001:** Enrichment factor (EF) of proteins isolated by MXP compared to the input lysate. The EF for each protein was calculated by dividing the average NSAF value across three MXP-pulldown replicates by the average NSAF value of three input lysate replicates. Proteins that were not detected in the input samples, but were isolated by MXP, were defined as “enriched”.

	Enrichment Factor (EF)	
	BRCA1−	BRCA1+
TOP2A	1.92	1.01
MDC1	Enriched	Enriched
RFC1	5.25	10.60
RFC4	3.61	0.80
NSD2	11.05	Enriched
RIF1	0.76	0.58

## Data Availability

The mass spectrometry proteomics data have been deposited to the ProteomeXchange Consortium via the PRIDE partner repository with the dataset identifiers PXD069034 and PXD069401. Reviewer access details: Log in to the PRIDE [84] website using the following details: Project accession: PXD069034, Token: dvq30Oz6UvZ6. Project accession: PXD069401, Token: itbudLh5UMbS. Additional data will be made available upon request.

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
