# Peer review of "Surveying the Proteome-Wide Landscape of Mitoxantrone and Examining Drug Sensitivity in BRCA1-Deficient Ovarian Cancer Using Quantitative Proteomics"

_proteomes, 2025, doi:10.3390/proteomes13040061_

Round 1

Reviewer 1 Report

Comments and Suggestions for Authors

Wallin et al. systematically studied the protein abundance changes in UWB1.289 ovarian cancer cells with or without BRCA1 after treatment with Mitoxantrone (MX), as well as the potential targets of MX. Through comprehensive data analysis, the authors provide insights into MX sensitivity in BRCA1-deficient cancer. Overall, this study is well executed and represents a general approach to gaining insights into the pharmacology of small-molecule drugs via chemical proteomics. I would support publication if the authors can address the following questions:

  1. In Figure 2A, can the authors explain the IC₅₀ of MX in BRCA1⁺ cells, given that the highest concentration tested is 2.5 µM, while the IC₅₀ (reported as EC₅₀ in the paper) is presented as 3 µM?
  2. How do the authors interpret the direct versus indirect targets of MX in Figure 4, since the proteins pulled down by streptavidin are not necessarily direct targets?

Author Response

Wallin et al. systematically studied the protein abundance changes in UWB1.289 ovarian cancer cells with or without BRCA1 after treatment with Mitoxantrone (MX), as well as the potential targets of MX. Through comprehensive data analysis, the authors provide insights into MX sensitivity in BRCA1-deficient cancer. Overall, this study is well executed and represents a general approach to gaining insights into the pharmacology of small-molecule drugs via chemical proteomics. I would support publication if the authors can address the following questions:

Comment 1: In Figure 2A, can the authors explain the IC₅₀ of MX in BRCA1⁺ cells, given that the highest concentration tested is 2.5 µM, while the IC₅₀ (reported as EC₅₀ in the paper) is presented as 3 µM?

Response 1: Thank you for pointing this error out! We have modified Figure 2A to include the proper estimated EC50 value, which is greater than > 2.5 μM. This change can be found on page 7, after line 276.

Comment 2: How do the authors interpret the direct versus indirect targets of MX in Figure 4, since the proteins pulled down by streptavidin are not necessarily direct targets?

Response 2: We appreciate this comment, as it is important to acknowledge the limitations of the affinity purification approach using MXP. Affinity purification techniques, such as our MXP affinity purification, cannot definitively distinguish between direct from indirect protein-protein interactions. From the STRING analysis in Figure 5, we observe clusters of functionally related proteins with documented physical interactions, such as the proteins involved in rRNA metabolism or mitochondrial translation. This is evidence that MXP is isolating known protein complexes, but the approach does not delineate which protein(s) in the complex MXP directly physically interacts. However, the results provide insight into pathways that are targeted by MXP through the complexes that are isolated. We have added additional explanation regarding this on page 14, lines 435-442.

Reviewer 2 Report

Comments and Suggestions for Authors

The manuscript authored by Wallin et. al. investigates Mitoxantrone (MX) activity in BRCA1- and BRCA1+ UWB1.289 ovarian cancer cells using mass spectrometry (MS)-based proteomics and affinity-purification coupled to MS analysis to derive MX targets. Although previous studies have addressed MX targets at the cellular level, the topic is of interest, considering MX use in oncology. The article is well written and could be considered for publication following some minor improvements:

Major:

  1. The manuscript mentions under multiple paragraphs that “log2 protein abundance ratio was calculated as the ratio of the summed abundances of identified proteins”. I don’t understand this: how did the authors translated the quan values from peptides (which is actually what they measured using their approach) to proteins and then to proteins groups. Please explain and modify accordingly in the manuscript.
  2. The authors mention that the trypsin digestion was performed for 4 hours. This is unusual considering the kinetic of trypsin cleavage. Did the authors used a special recombinant enzyme with faster kinetics? Please provide graphs in the supplemental section regarding missed cleavage distribution for all the samples.

Minor:

  1. For proteomics analysis the authors mention reverse-phase C18 chromatography fro fractionation and then analysis of each fraction on a C18 Waters CSH column. What is the difference between the first and second dimension of C18 analysis? Please mention also in the manuscript accordingly.
  2. Figure 2D: Please add the title for the second volcano plot.
  3. What was the dynamic range of the quantification using TMT? Please add a corresponding plot in the supplemental section.
  4. Supplemental Figure S3A: Please add the electropherogram or the lane view of the obtained results.

Author Response

The manuscript authored by Wallin et. al. investigates Mitoxantrone (MX) activity in BRCA1- and BRCA1+ UWB1.289 ovarian cancer cells using mass spectrometry (MS)-based proteomics and affinity-purification coupled to MS analysis to derive MX targets. Although previous studies have addressed MX targets at the cellular level, the topic is of interest, considering MX use in oncology. The article is well written and could be considered for publication following some minor improvements:

Major:

Comment 1: The manuscript mentions under multiple paragraphs that “log2 protein abundance ratio was calculated as the ratio of the summed abundances of identified proteins”. I don’t understand this: how did the authors translated the quan values from peptides (which is actually what they measured using their approach) to proteins and then to proteins groups. Please explain and modify accordingly in the manuscript.

Response 1: Thank you for this comment, we agree that this description should be re-written. The log2 protein abundance ratio was calculated by dividing the summed protein abundance quantified across four replicates for one treatment group by the summed protein abundance of another treatment group. Treatment groups (MX vs DMSO) for either BRCA1- or BRCA1+ UWB1.289 samples were compared. The associated adjusted p-values were determined by analysis of variance (ANOVA) using the Benjamini-Hochberg method. This change can be found on page 4, lines 139-144.

Comment 2: The authors mention that the trypsin digestion was performed for 4 hours. This is unusual considering the kinetic of trypsin cleavage. Did the authors used a special recombinant enzyme with faster kinetics? Please provide graphs in the supplemental section regarding missed cleavage distribution for all the samples.

Response 2: Thank you for pointing this out. To clarify, after trypsin digestion, the digested samples were analyzed by using nano-LC-MS and had > 93% zero missed cleavages for all of the samples, making them suitable for TMT labeling and analysis by mass spectrometry. This is the standard quality assurance protocol for the core facility. The samples met the criteria for efficient digestion, which is necessary for the subsequent steps of mass spectrometry analysis. This clarification was made to the methods on page 4, lines 122-124. While we appreciate the suggestion to add the cleavage distribution in the supplement, the requested figure would not change or add to the scope of the paper.

Minor:

Comment 1: For proteomics analysis the authors mention reverse-phase C18 chromatography for fractionation and then analysis of each fraction on a C18 Waters CSH column. What is the difference between the first and second dimension of C18 analysis? Please mention also in the manuscript accordingly.

Response 1: The first reverse-phase C18 fractionation of peptides was done at a high pH (pH 10). The second reverse-phase C18 fractionation was performed at low pH in a buffer containing 0.05% TFA. We have clarified this in the manuscript which can be found on page 4, lines 126-136.

Comment 2: Figure 2D: Please add the title for the second volcano plot.

Response 2: We appreciate the Reviewer’s attention to detail! The title for the second volcano plot was added and can be found on page 7, after line 276.

Comment 3: What was the dynamic range of the quantification using TMT? Please add a corresponding plot in the supplemental section.

Response 3: We appreciate this comment and have added an additional figure to the Supplement (Supplementary Figure S1), which addresses the dynamic range observed in the TMT experiment (Supplement-page 2, lines 17-21). The dynamic range was 3027.9x.